# Asymptotic optimality of adaptive importance sampling

**Bernard Delyon**
IRMAR
University of Rennes 1
bernard.delyon@univ-rennes1.fr

**François Portier**
Télécom ParisTech
University of Paris-Saclay
francois.portier@gmail.com

## Abstract

*Adaptive importance sampling* (AIS) uses past samples to update the *sampling policy* $q_t$. Each stage $t$ is formed with two steps : (i) to explore the space with $n_t$ points according to $q_t$ and (ii) to exploit the current amount of information to update the sampling policy. The very fundamental question raised in this paper concerns the behavior of empirical sums based on AIS. Without making any assumption on the *allocation policy* $n_t$, the theory developed involves no restriction on the split of computational resources between the explore (i) and the exploit (ii) step. It is shown that AIS is asymptotically optimal : the asymptotic behavior of AIS is the same as some "oracle" strategy that knows the targeted sampling policy from the beginning. From a practical perspective, weighted AIS is introduced, a new method that allows to forget poor samples from early stages.

## 1 Introduction

The adaptive choice of a sampling policy lies at the heart of many fields of *Machine Learning* where former Monte Carlo experiments guide the forthcoming ones. This includes for instance *reinforcment learning* [19, 27, 30] where the optimal policy maximizes the reward; inference in *Bayesian* [6] or *graphical models* [21]; *optimization* based on stochastic gradient descent [34] or without using the gradient [18]; *rejection sampling* [12]. *Adaptive importance sampling* (AIS) [25, 2], which extends the basic Monte Carlo integration approach, offers a natural probabilistic framework to describe the evolution of sampling policies. The present paper establishes, under fairly reasonable conditions, that AIS is asymptotically optimal, i.e., learning the sampling policy has no cost asymptotically.

Suppose we are interested in computing some integral value $\int \varphi$, where $\varphi : \mathbb{R}^d \to \mathbb{R}$ is called the integrand. The importance sampling estimate of $\int \varphi$ based on the sampling policy $q$, is given by

$$n^{-1} \sum_{i=1}^n \frac{\varphi(x_i)}{q(x_i)}, \tag{1}$$

where $(x_1, \ldots x_n) \overset{\text{i.i.d.}}{\sim} q$. The previous estimate is unbiased. It is well known, e.g., [16, 13], that the optimal sampling policy, regarding the variance, is when $q$ is proportional to $|\varphi|$. A slightly different context where importance sampling still applies is Bayesian estimation. Here the targeted quantity is $\int \varphi \pi$ and we only have access to an unnormalized version $\pi_u$ of the density $\pi = \pi_u / \int \pi_u$. Estimators usually employed are

$$\sum_{i=1}^n \frac{\varphi(x_i)\pi_u(x_i)}{q(x_i)} \left/ \sum_{i=1}^n \frac{\pi_u(x_i)}{q(x_i)} \right. . \tag{2}$$

In this case, the optimal sampling policy $q$ is proportional to $|\varphi - \int \varphi\pi|\pi$ (see [9] or section B.3 in the supplementary material).

Because appropriate policies naturally depend on $\varphi$ or $\pi$, we generally cannot simulate from them. They are then approximated adaptively, by densities from which we can simulate, using the information gathered from the past stages. This is the very spirit of AIS. At each stage $t$, the value $I_t$, standing for the current estimate, is updated using i.i.d. new samples $x_{t,1}, \ldots x_{t,n_t}$ from $q_t$, where $q_t$ is a probability density function that might depend on the past stages $1, \ldots t-1$. The distribution $q_t$, called the *sampling policy*, targets some optimal, at least suitable, sampling policy. The sequence $(n_t) \subset \mathbb{N}^*$, called the *allocation policy*, contains the number of particles generated at each stage.

The following algorithm describes the AIS schemes for the classical integration problem. For the Bayesian problem, it suffices to change the estimate according to (2). This is a generic representation of AIS as no explicit update rule is specified (this will be discussed just below).

**Algorithm 1** (AIS).
**Inputs**: *The number of stages $T \in \mathbb{N}^*$, the allocation policy $(n_t)_{t=1,\ldots T} \subset \mathbb{N}^*$, the sampler update procedure, the initial density $q_0$.*

---

*Set $S_0 = 0$, $N_0 = 0$. For $t$ in $1, \ldots T$ :*

(i) *(Explore) Generate $(x_{t,1}, \ldots x_{t,n_t})$ from $q_{t-1}$*

(ii) *(Exploit)*

    (a) *Update the estimate:*
$$S_t = S_{t-1} + \sum_{i=1}^{n_t} \frac{\varphi(x_{t,i})}{q_{t-1}(x_{t,i})}$$
$$N_t = N_{t-1} + n_t$$
$$I_t = N_t^{-1} S_t$$

    (b) *Update the sampler $q_t$*

---

Pioneer works on adaptive schemes include [20] where, within a two-stages procedure, the sampling policy is chosen out of a parametric family; this is further formalized in [14]; [25] introduces the idea of a multi-stages approach where all the previous stages are used to update the sampling policy (see also [29] regarding the choice of the loss function); [26] investigates the use of control variates coupled with importance sampling; the *population Monte Carlo* approach [3, 2] offers a general framework for AIS and has been further studied using parametric mixtures [8, 9]; see also [5, 32] for a variant called *multiple adaptive importance sampling*; see [11] for a recent review. In [33, 23], using kernel smoothing, *nonparametric importance sampling* is introduced. The approach of choosing $q_t$ out of a parametric family should also be contrasted with the non parametric approach based on particles often refereed to as *sequential Monte Carlo* [6, 4, 10] whose context is different as traditionally the targeted distribution changes with $t$. The distribution $q_{t-1}$ is then a weighted sum of Dirac masses $\sum_i w_{t-1,i} \delta_{x_{t-1,i}}$, and updating $q_t$ follows from adjustment of the weights.

The theoretical properties of adaptive schemes are difficult to derive due to the recycling of the past samples at each stage and hence to the lack of independence between samples. Among the update based on a parametric family, the convergence properties of the Kullback-Leibler divergence between the estimated and the targeted distribution are studied in [8]. Properties related to the asymptotic variance are given in [9]. Among nonparametric update, [33] establishes fast convergence rates in a two-stages strategy where the number of samples used in each stage goes to infinity. For sequential Monte Carlo, limit theorems are given for instance in [6, 4, 10]. All these results are obtained when $T$ is fixed and $n_T \to \infty$ and therefore misses the true nature of the adaptive schemes for which the asymptotic should be made with respect to $T$.

Recently, a more realistic asymptotic regime was considered in [22] in which the allocation policy $(n_t)$ is a fixed growing sequence of integers. The authors establish the consistency of the estimate when the update is conducted with respect to a parametric family but depends *only* on the last stage. They focus on multiple adaptive importance sampling [5, 32] which is different than AIS (see Remark 2 below for more details).

In this paper, folllowing the same spirit as [8, 9, 2], we study *parametric* AIS as presented in the AIS algorithm when the policy is chosen out of a parametric family of probability density functions. Our analysis focuses on the following 3 key points which are new to the best of our knowledge.

- A central limit theorem is established for the AIS estimate $I_t$. It involves high-level conditions on the sampling policy estimate $q_t$ (which will be easily satisfied for parametric updates). Based on the martingale property associated to some sequences of interest, the asymptotic is not with $T$ fixed and $n_T \to \infty$, but with the number of samples $n_1 + \cdots + n_T \to \infty$. In particular, the allocation policy $(n_t)$ is not required to grow to infinity. This is presented in section 2.

- The high-level conditions are verified in the case of parametric sampling policies with updates taking place in a general framework inspired by the paradigm of empirical risk minimization (several concrete examples are provided). This establishes the asymptotic optimality of AIS in the sense that the rate and the asymptotic variance coincide with some "oracle" procedure where the targeted policy is known from the beginning. The details are given in section 3.

- A new method, called weighted AIS (wAIS) is designed in section 4 to eventually forget bad samples drawn during the early stages of AIS. Our numerical experiments shows that (i) wAIS accelerates significantly the convergence of AIS and (ii) small allocation policies $(n_t)$ (implying more frequent updates) give better results than large $(n_t)$ (at equal number of requests to $\varphi$). This last point supports empirically the theoretical framework adopted in the paper.

All the proofs are given in the supplementary material.

## 2 Central limit theorems for AIS

The aim of the section is to provide conditions on the sampling policy $(q_t)$ under which a central limit theorem holds for AIS and normalized AIS.

For the sake of generality and because it will be useful in the treatment of normalized estimators, we consider the multivariate case where $\varphi = (\varphi_1, \ldots \varphi_p) : \mathbb{R}^d \to \mathbb{R}^p$. In the whole paper, $\int \varphi$ is with respect to the Lebesgue measure, $\| \cdot \|$ is the Euclidean norm, $\mathcal{I}_p$ is the identity matrix of size $(p, p)$.

To study the AIS algorithm, it is appropriate to work at the sample time scale as described below rather than at the sampling policy scale as described in the introduction. The sample $x_{t,i}$ (resp. the policy $q_t$) of the previous section ($t$ is the block index and $i$ the sample index within the block) is now simply denoted $x_j$ (resp. $q_j$), where $j = n_1 + \ldots n_t + i$ is the sample index in the whole sequence $1, \ldots n$, with $n = N_T$. The following algorithm is the same as Algorithm 1 (no explicit update rule is provided) but is expressed at the sample scale.

**Algorithm 2** (AIS at sample scale)**.**
***Inputs****: The number of stages $T \in \mathbb{N}^*$, the allocation policy $(n_t)_{t=1,\ldots T} \subset \mathbb{N}^*$, the sampler update procedure, the initial density $q_0$.*

---

*Set $S_0 = 0$. For $j$ in $1, \ldots n$ :*

*(i) (Explore) Generate $x_j$ from $q_{j-1}$*

*(ii) (Exploit)*

    *(a) Update the estimate:*
$$S_j = S_{j-1} + \frac{\varphi(x_j)}{q_{j-1}(x_j)}$$
$$I_j = j^{-1} S_j$$
    *(b) Update the sampler $q_j$ whenever $j \in \{N_t = \sum_{s=1}^{t} n_s : t \geqslant 1\}$*

---

### 2.1 The martingale property

Define $\Delta_j$ as the $j$-th centered contribution to the sum $S_j$: $\Delta_j = \varphi(x_j)/q_{j-1}(x_j) - \int \varphi$. Define, for all $n \geqslant 1$,

$$M_n = \sum_{j=1}^{n} \Delta_j.$$

The filtration we consider is given by $\mathscr{F}_n = \sigma(x_1, \dots x_n)$. The quadratic variation of $M$ is given by $\langle M \rangle_n = \sum_{j=1}^n \mathbb{E}\big[\Delta_j \Delta_j^T \mid \mathscr{F}_{j-1}\big]$. Set

$$V(q, \varphi) = \int \frac{\big(\varphi(x) - q(x) \int \varphi\big) \big(\varphi(x) - q(x) \int \varphi\big)^T}{q(x)} dx. \tag{3}$$

**Lemma 1.** *Assume that for all $1 \leqslant j \leqslant n$, the support of $q_j$ contains the support of $\varphi$, then the sequence $(M_n, \mathscr{F}_n)$ is a martingale. In particular, $I_n$ is an unbiased estimate of $\int \varphi$. In addition, the quadratic variation of $M$ satisfies $\langle M \rangle_n = \sum_{j=1}^n V(q_{j-1}, \varphi)$.*

## 2.2 A central limit theorem for AIS

The following theorem describes the asymptotic behavior of AIS. The conditions will be verified for parametric updates in section 3 (see Theorem 3) in which case the asymptotic variance $V_*$ will be explicitly given.

**Theorem 1** (central limit theorem for AIS). *Assume that the sequence $q_n$ satisfies*

$$V(q_n, \varphi) \to V_*, \qquad a.s. \tag{4}$$

*for some $V_* \geqslant 0$ and that there exists $\eta > 0$ such that*

$$\sup_{j \in \mathbb{N}} \int \frac{\|\varphi\|^{2+\eta}}{q_j^{1+\eta}} < \infty, \qquad a.s. \tag{5}$$

*Then we have*

$$\sqrt{n}\left(I_n - \int \varphi\right) \xrightarrow{\text{d}} \mathcal{N}(0, V_*).$$

**Remark 1** (zero-variance estimate). *Suppose that $p = 1$ (recalling that $\varphi : \mathbb{R}^d \to \mathbb{R}^p$). Theorem 1 includes the degenerate case $V_* = 0$. This happens when the integrand has constant sign and the sampling policy is well chosen, i.e. $q_n \to |\varphi|/\int |\varphi|$. In this case, we have that $\sqrt{n}(I_n - \int \varphi) = o_p(1)$, meaning that the standard Monte Carlo convergence rate $(1/\sqrt{n})$ has been improved. This is inline with the results presented in [33] where fast rates of convergence (compared to standard Monte Carlo) are obtained under restrictive conditions on the allocation policy $(n_t)$. Note that other techniques such as control variates, kernel smoothing or Gaussian quadrature can achieve fast convergence rates [24, 28, 7, 1].*

**Remark 2** (adaptive multiple importance sampling). *Another way to compute the importance weights, called multiple adaptive importance sampling, has been introduced in [32] and has been successfully used in [26, 5]. This consists in replacing $q_{j-1}$ in the computation of $S_j$ by $\bar{q}_{j-1} = \sum_{i=1}^j q_{i-1}/j$, $x_j$ still being drawn under $q_{j-1}$. The intuition is that this averaging will reduce the effect of exceptional points $x_j$ for which $|\varphi(x_j)| \gg q_{j-1}(x_j)$ (but $|\varphi(x_j)| \not\gg \bar{q}_{j-1}(x_j)$). Our approach is not able to study this variant, simply because the martingale property described previously is not anymore satisfied.*

## 2.3 Normalized AIS

The normalization technique described in (2) is designed to compute $\int \varphi \pi$, where $\pi$ is a density. It is useful in the Bayesian context where $\pi$ is only known up to a constant. As this technique seems to provide substantial improvements compared to unnormalized estimates (i.e., (1) with $\varphi$ replaced by $\varphi \pi$), we recommend to use it even when the normalized constant of $\pi$ is known. Normalized estimators are given by

$$I_n^{(\text{norm})} = \frac{I_n(\varphi \pi)}{I_n(\pi)}, \qquad \text{with} \quad I_n(\psi) = n^{-1} \sum_{j=1}^n \psi(x_j)/q_{j-1}(x_j).$$

Interestingly, normalized estimators are weighted least-squares estimates as they minimize the function $a \mapsto \sum_{j=1}^n (\pi(x_j)/q_{j-1}(x_j))(\varphi(x_j) - a)^2$. In contrast with $I_n$, $I_n^{(\text{norm})}$ has the following shift-invariance property : whenever $\varphi$ is shifted by $\mu$, $I_n^{(\text{norm})}$ simply becomes $I_n^{(\text{norm})} + \mu$. Because $I_n(\varphi \pi)$ and $I_n(\pi)$ are of the same kind as $I_n$ defined in the second AIS algorithm, a straightforward application of Theorem 1 (with $(\varphi^T \pi, \pi)^T$ in place of $\varphi$).

**Corollary 1** (central limit theorem for normalized AIS). *Suppose that (4) and (5) hold with $(\varphi^T \pi, \pi)^T$ (in place of $\varphi$). Then we have*

$$\sqrt{n}\left(I_n^{(norm)} - \int \varphi \pi\right) \overset{\mathrm{d}}{\to} \mathcal{N}(0, UV_*U^T),$$

*with $U = (\mathcal{I}_p, -\int \varphi \pi)$.*

## 3 Parametric sampling policy

From this point forward, the sampling policies $q_t$, $t = 1, \dots T$ (we are back again to the sampling policy scale as in Algorithm 1), are chosen out of a parametric family of probability density functions $\{q_\theta : \theta \in \Theta\}$. All our examples fit the general framework of empirical risk minimization over the parameter space $\Theta \subset \mathbb{R}^q$, where $\theta_t$ is given by

$$\theta_t \in \operatorname{argmin}_{\theta \in \Theta} R_t(\theta), \tag{6}$$

$$R_t(\theta) = \sum_{s=1}^{t} \sum_{i=1}^{n_s} \frac{m_\theta(x_{s,i})}{q_{s-1}(x_{s,i})},$$

where $q_s$ is a shortcut for $q_{\theta_s}$, $m_\theta : \mathbb{R}^d \to \mathbb{R}$ might be understood as a loss function (see the next section for examples). Note that $R_t/N_t$ is an unbiased estimate of the risk $r(\theta) = \int m_\theta$.

### 3.1 Examples of sampling policy

We start by introducing a particular case, which is one of the simplest way to implement AIS. Then we will provide more general approaches. In what follows, the targeted policy, denoted by $f$, is chosen by the user and represents the distribution from which we wish to sample. It often reflects some prior knowledge on the problem of interest. If $\varphi : \mathbb{R}^d \to \mathbb{R}^p$, with $p = 1$, then (as discussed in the introduction) $f \propto |\varphi|$ is optimal for (1) and $f \propto |\varphi - \int \varphi \pi|\pi$ is optimal for (2). In the Bayesian context where many integrals $\int(\varphi_1, \dots \varphi_p)d\pi$ need to be computed, a usual choice is $f = \pi$. All the following methods only require calls to an unnormalized version of $f$.

**Method of moments with Student distributions.** In this case $(q_\theta)_{\theta \in \Theta}$ is just the family of multivariate Student distributions with $\nu > 2$ degrees of freedom (fixed parameter). The parameter $\theta$ contains a location and a scale parameter $\mu$ and $\Sigma$. This family has two advantages: the parameter $\nu$ allows tuning for heavy tails, and estimation is easy because moments of $q_\theta$ are explicitly related to $\theta$. A simple unbiased estimate for $\mu$ is $(1/N_t) \sum_{s=1}^{t} \sum_{i=1}^{n_s} x_{s,i} f(x_{s,i})/q_{s-1}(x_{s,i})$, but, as mentioned in section 2.3, we prefer to use the normalized estimate (using the shortcut $q_s$ for $q_{\theta_s}$):

$$\mu_t = \sum_{s=1}^{t} \sum_{i=1}^{n_s} x_{s,i} \frac{f(x_{s,i})}{q_{s-1}(x_{s,i})} \bigg/ \sum_{s=1}^{t} \sum_{i=1}^{n_s} \frac{f(x_{s,i})}{q_{s-1}(x_{s,i})}, \tag{7}$$

$$\Sigma_t = \left(\frac{\nu - 2}{\nu}\right) \sum_{s=1}^{t} \sum_{i=1}^{n_s} (x_{s,i} - \mu_t)(x_{s,i} - \mu_t)^T \frac{f(x_{s,i})}{q_{s-1}(x_{s,i})} \bigg/ \sum_{s=1}^{t} \sum_{i=1}^{n_s} \frac{f(x_{s,i})}{q_{s-1}(x_{s,i})}. \tag{8}$$

**Generalized method of moments (GMM).** This approach includes the previous example. The policy is chosen according to a moment matching condition, i.e., $\int g q_\theta = \int g f$ for some function $g : \mathbb{R}^d \to \mathbb{R}^D$. For instance, $g$ might be given by $x \mapsto x$ or $x \mapsto xx^T$ (both are considered in the Student case). Following [17], choosing $\theta$ such that the empirical moments of $g$ coincide with $\int g q_\theta$ might be impossible. We rather compute $\theta_t$ as the minimum of

$$\left\| \mathbb{E}_\theta(g) - \left(\sum_{s=1}^{t} \sum_{i=1}^{n_s} g(x_{s,i}) \frac{f(x_{s,i})}{q_{s-1}(x_{s,i})} \bigg/ \sum_{s=1}^{t} \sum_{i=1}^{n_s} \frac{f(x_{s,i})}{q_{s-1}(x_{s,i})}\right) \right\|^2.$$

Equivalently,

$$\theta_t \in \operatorname{argmin}_{\theta \in \Theta} \sum_{s=1}^{t} \sum_{i=1}^{n_s} \|\mathbb{E}_\theta(g) - g(x_{s,i})\|^2 \frac{f(x_{s,i})}{q_{s-1}(x_{s,i})},$$

which embraces the form given by (6), with $m_\theta = \|\mathbb{E}_\theta(g) - g\|^2 f$.

**Kullback-Leibler approach.** Following [31, section 5.5], define the Kullback-Leibler risk as $r(\theta) = -\int \log(q_\theta)f$. Update of $\theta_t$ is done by minimizing the current estimator of $N_t r(\theta)$ given by

$$R_t(\theta) = R_{t-1}(\theta) - \sum_{i=1}^{n_t} \frac{\log(q_\theta(x_{t,i}))f(x_{t,i})}{q_{t-1}(x_{t,i})}. \tag{9}$$

**Variance approach.** Another approach, when $\varphi : \mathbb{R}^d \to \mathbb{R}^p$ with $p = 1$, consists in minimizing the variance over the class of sampling policies. In this case, define $r(\theta) = \int \varphi^2/q_\theta$, and follow a similar approach as before by minimizing at each stage,

$$R_t(\theta) = R_{t-1}(\theta) + \sum_{i=1}^{n_t} \frac{\varphi(x_{t,i})^2}{q_\theta(x_{t,i})q_{t-1}(x_{t,i})}. \tag{10}$$

This case represents a different situation than the Kullback-Leibler approach and the GMM. Here, the sampling policy is selected optimally with respect to a particular function $\varphi$ whereas for KL and GMM the sampling policy is driven by a targeted distribution $f$.

**Remark 3** (computation cost). *The update rule (6) might be computationally costly but alternatives exist. For instance, when $q_\theta$ is a family of Gaussian distributions, closed formulas are available for (10). In fact we are in the case of weighted maximum likelihood estimation for which we find exactly (7) and (8), with $\nu = \infty$. This is computed online at no cost. Another strategy to reduce the computation time is to use online stochastic gradient descent in (6).*

**Remark 4** (block estimator). *In [22], the authors suggest to update $\theta$ based only on the particles from the last stage. For the Kullback-Leibler update, (9) would be replaced by $R_t(\theta) = -\sum_{i=1}^{n_t} \log(q_\theta(x_{t,i}))f(x_{t,i})/q_{t-1}(x_{t,i})$. While this update makes easier the theoretical analysis (assuming that $n_t \to \infty$), its main drawback is that most of the computing effort is forgotten at each stage as the previous computations are not used.*

### 3.2 Consistency of the sampling policy and asymptotic optimality of AIS

The updates described before using GMM, the Kullback-Leibler divergence or the variance, all fit within the framework of empirical risk minimization, given by (6), which rewritten at the sample scale gives

$$
\begin{aligned}
R_j(\theta) &= R_{j-1}(\theta) + \frac{m_\theta(x_j)}{q_{j-1}(x_j)} \\
&- \text{if } j \in \{N_t : t \geqslant 1\} \text{ then :} \qquad \theta_j \in \operatorname{argmin}_{\theta \in \Theta} R_j(\theta) \\
&\qquad\qquad\qquad\qquad\qquad\qquad q_j = q_{\theta_j} \\
&- \text{else :} \qquad\qquad\qquad\qquad\qquad q_j = q_{j-1}.
\end{aligned}
$$

The proof follows from a standard approach from $M$-estimation theory [31, Theorem 5.7] but a particular attention shall be payed to the uniform law of large numbers because of the missing i.i.d. property of the sequences of interest.

**Theorem 2** (concistency of the sampling policy). *Set $M(x) = \sup_{\theta \in \Theta} m_\theta(x)$. Assume that $\Theta \subset \mathbb{R}^q$ is a compact set and that*

$$\int M(x)dx < \infty, \quad \sup_{\theta \in \Theta} \int \frac{M(x)^2}{q_\theta(x)}dx < \infty, \quad and \quad \forall \theta \neq \theta_*, \ r(\theta) = \int m_\theta > \int m_{\theta_*}. \tag{11}$$

*If moreover, for any $x \in \mathbb{R}^d$, the function $\theta \mapsto m_\theta(x)$ is continuous on $\mathbb{R}^q$, then*

$$\theta_n \to \theta_*, \qquad a.s.$$

The conclusion given in Theorem 2 permits to check the conditions of Theorem 1. This leads to the following result.

**Theorem 3** (asymptotic optimality of AIS). *Under the assumptions of Theorem 2, if there exists $\eta > 0$ such that $\sup_{\theta \in \Theta} \int \|\varphi\|^{2+\eta}/q_\theta^{1+\eta} < \infty$, we have*

$$\sqrt{n}\left(I_n - \int \varphi\right) \xrightarrow{\text{d}} \mathcal{N}\left(0, V(q_{\theta_*}, \varphi)\right),$$

*where $V(\cdot, \cdot)$ is defined in Equation (3).*

**Remark 5** (the oracle property). *From (11), we deduce that $q_{\theta_*}$ is the unique minimizer of the risk function $r$. The risk function based on GMM or the Kullback-Leibler approach (described in section 3.1) is derived from a certain targeted density $f$ in such a way that if $q_\theta = f$, then $r(\theta)$ is a minimum. Hence under the identifiability conditions of Theorem 2, if in addition $f \in \{q_\theta : \theta \in \Theta\}$, we have that $q_{\theta_*} = f$. This means that asymptotically, AIS achives the same variance as the "oracle" importance sampling method based on the (fixed) sampler $f$.*

**Corollary 2** (asymptotic optimality for normalized AIS). *Under the assumptions of Theorem 2, if there exists $\eta > 0$ such that $\sup_{\theta \in \Theta} \int \|(\varphi^T \pi, \pi)\|^{2+\eta}/q_\theta^{1+\eta} < \infty$, we have*

$$\sqrt{n}\left(I_n^{(norm)} - \int \varphi\pi\right) \xrightarrow{\mathrm{d}} \mathcal{N}\left(0, UV(q_{\theta_*}, (\varphi^T\pi, \pi)^T)U^T\right),$$

*with $U$ defined in Corollary 1 and $V(\cdot, \cdot)$ defined in Equation (3).*

## 4 Weighted AIS

We follow ideas from [9, section 4] to develop a novel method to estimate $\int \varphi\pi$. The method is called weighted adaptive importance sampling (wAIS), and will automatically re-weights each sample depending on its accuracy. It allows in practice to forget poor samples generated during the early stages. For clarity, suppose that $\varphi : \mathbb{R}^d \to \mathbb{R}^p$ with $p = 1$. Define the weighted estimate, for any function $\psi$,

$$I_T^{(\alpha)}(\psi) = N_T^{-1} \sum_{t=1}^T \alpha_{T,t} \sum_{i=1}^{n_t} \frac{\psi(x_{t,i})}{q_{t-1}(x_{t,i})}.$$

Note that for any sequence $(\alpha_{T,1}, \dots \alpha_{T,T})$ such that $\sum_{t=1}^T n_t \alpha_{T,t} = N_T$, $I_T^{(\alpha)}(\psi)$ is an unbiased estimate of $\int \psi$. Let $\sigma_t^2 = \mathbb{E}[V(q_{t-1}, \varphi)]$ where $V(\cdot, \cdot)$ is defined in Equation (3). The variance of $I_T^{(\alpha)}(\varphi)$ is $N_T^{-2} \sum_{t=1}^T \alpha_{T,t}^2 n_t \sigma_t^2$ which minimized w.r.t. $(\alpha)$ gives $\alpha_{T,t} \propto \sigma_t^{-2}$, for each $t = 1, \dots T$. In [9], a re-weighting is proposed using estimates of $\sigma_t$ (based on sample of the $t$-th stage). We propose the following weights

$$\alpha_{T,t}^{-1} \propto \sum_{i=1}^{n_t} \left(\frac{\pi(x_{t,i})}{q_{t-1}(x_{t,i})} - 1\right)^2, \tag{12}$$

satisfying the constraints $\sum_{t=1}^T n_t \alpha_{T,t} = N_T$. The wAIS estimate is the (weighted and normalized) AIS estimate given by

$$I_T^{(\alpha)}(\varphi\pi)/I_T^{(\alpha)}(\pi). \tag{13}$$

In contrast with the approach in [9], because our weights are based on the estimated variance of $\pi/q_{t-1}$, our proposal is free from the integrand $\varphi$ and thus reflects the overall quality of the $t$-th sample. This makes sense whenever many functions need to be integrated making inappropriate a re-weighting depending on a specific function. Another difference with [9] is that we use the true expectation, 1, in the estimate of the variance, rather than the estimate $(1/n_t)\sum_{i=1}^{n_t} \pi(x_{t,i})/q_{t-1}(x_{t,i})$. This permits to avoid the situation (common in high dimensional settings) where a poor sampler $q_{t-1}$ is such that $\pi(x_{t,i})/q_{t-1}(x_{t,i}) \simeq 0$, for all $i = 1, \dots n_t$, implying that the classical estimate of the variance is near 0, leading (unfortunately) to a large weight.

## 5 Numerical experiments

In this section, we study a toy Gaussian example to illustrate the practical behavior of AIS. Special interest is dedicated to the effect of the dimension $d$, the practical choice of $(n_t)$ and the gain given by wAIS introduced in the previous section. We set $N_T = 1e5$ and we consider $d = 2, 4, 8, 16$. The code is made available at `https://github.com/portierf/AIS`.

The aim is to compute $\mu_* = \int x\phi_{\mu_*,\sigma_*}(x)dx$ where $\phi_{\mu,\sigma} : \mathbb{R}^d \to \mathbb{R}$ is the probability density of $\mathcal{N}(\mu, \sigma^2\mathcal{I}_d)$, $\mu_* = (5, \dots 5)^T \in \mathbb{R}^d$, $\sigma_* = 1$. The sampling policy is taken in the collection of multivariate Student distributions of degree $\nu = 3$ denoted by $\{q_{\mu,\Sigma_0} : \mu \in \mathbb{R}^d\}$ with $\Sigma_0 =$

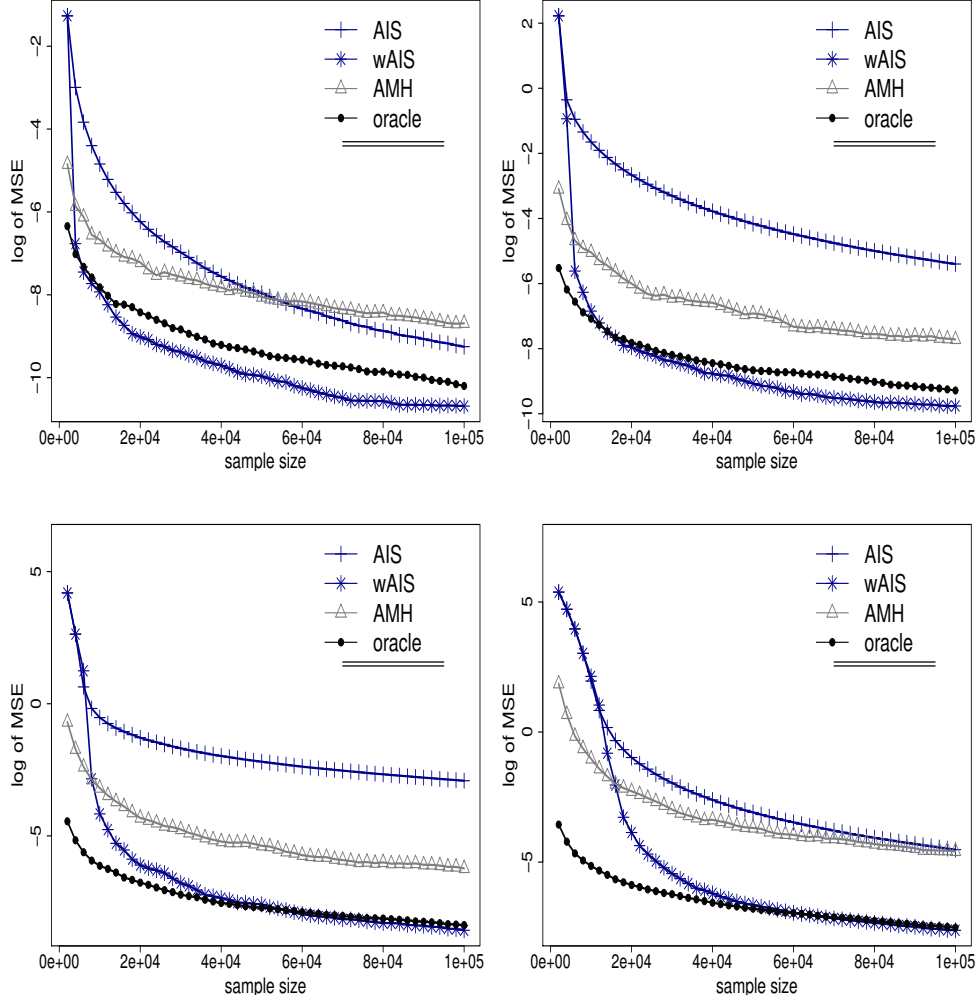

Figure 1: From left-to-right and top-to-bottom $d = 2, 4, 8, 16$. AIS and wAIS are computed with $T = 50$ with a constant allocation policy $n_t = 2e3$. Plotted is the logarithm of the MSE (computed for each method over 100 replicates) with respect to the number of requests to the integrand.

$\sigma_0 \mathcal{I}_d (\nu - 2)/\nu$ and $\sigma_0 = 5$. The initial sampling policy is set as $\mu_0 = (0, \dots 0) \in \mathbb{R}^d$. The mean $\mu_t$ is updated at each stage $t = 1, \dots T$ following the GMM approach as described in section 3, leading to the simple update formula

$$\mu_t = \sum_{s=1}^{t} \sum_{i=1}^{n_s} x_{s,i} \frac{f(x_{s,i})}{q_{s-1}(x_{s,i})} \Bigg/ \sum_{s=1}^{t} \sum_{i=1}^{n_s} \frac{f(x_{s,i})}{q_{s-1}(x_{s,i})} \ ,$$

with $f = \phi_{\mu_*, \sigma_*}$. In section C of the supplementary file, other results considering the update of the variance within the student family are provided.

As the results for the unnormalized approaches were far from being competitive with the normalized ones, we consider only normalized estimators. We also tried the weights proposed in [9] but the results were not competitive. The (normalized) AIS estimate of $\mu_*$ is simply given by $\mu_t$ as displayed above. The wAIS estimate of $\mu_*$ is computed using (13) with weights (12).

We also include the adaptive MH proposed in [15], where the proposal, assuming that $X_{i-1} = x$, is given by $\mathcal{N}\left(x, (2.4)^2 (C_i + \epsilon \mathcal{I}_d)/d\right)$, if $i > i_0$, and $\mathcal{N}(x, \mathcal{I}_d)$, if $i \leqslant i_0$, with $C_i$ the empirical covariance matrix of $(X_0, X_1, \dots X_{i-1})$, $i_0 = 1000$ and $\epsilon = 0.05$ (other configurations as for instance using only half of the chain have been tested without improving the results). Finally we

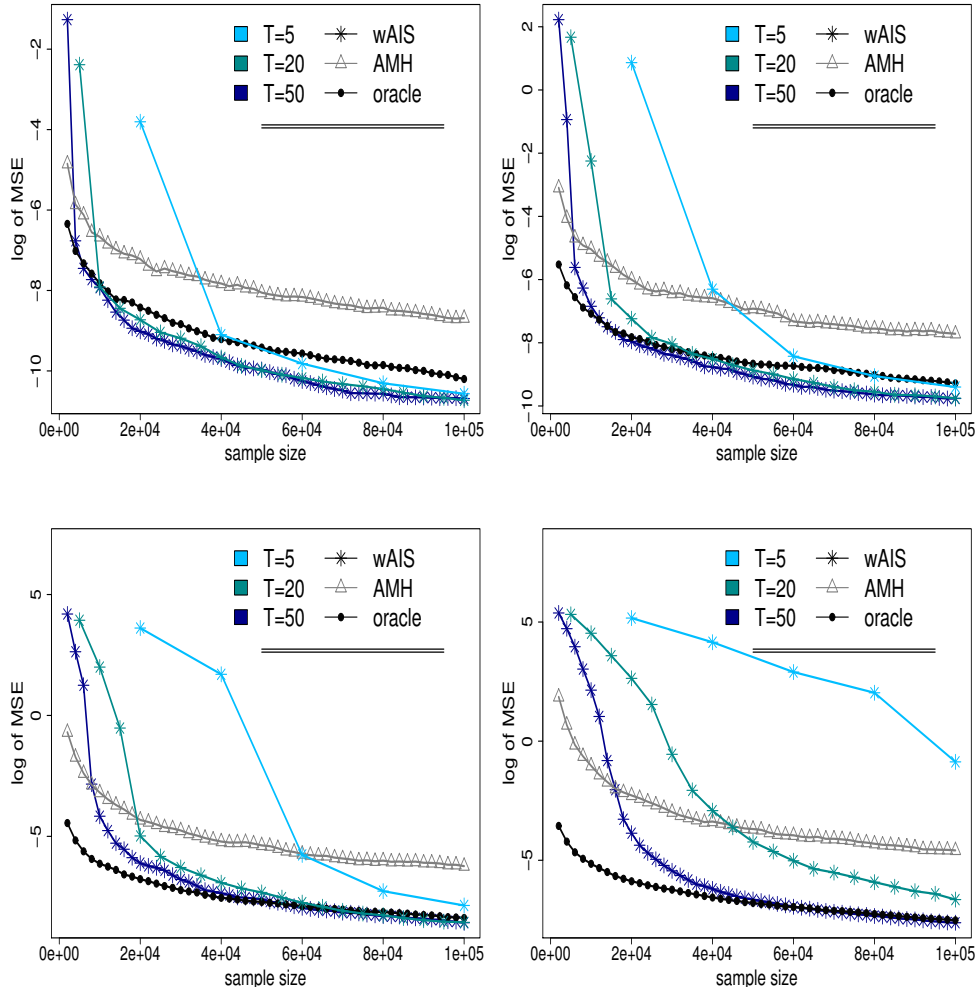

Figure 2: From left-to-right and top-to-bottom $d = 2, 4, 8, 16$. AIS and wAIS are computed with $T = 5, 20, 50$, each with a constant allocation policy, resp. $n_t = 2e4, 5e3, 2e3$. Plotted is the logarithm of the MSE (computed for each method over 100 replicates) with respect to the number of requests to the integrand.

consider a so called "oracle" method : importance sampling with fix policy $q_{\mu_*, \Sigma_*}$, with $\Sigma_* = \sigma_* \mathcal{I}_d (\nu - 2)/\nu$.

For each method that returns $\mu$, the mean squared error (MSE) is computed as the average of $\|\mu - \mu_*\|^2$ computed over 100 replicates of $\mu$.

In Figure 1, we compare the evolution of all the mentioned algorithms with respect to stages $t = 1, \dots T = 50$ with constant allocation policy $n_t = 2e3$ (for AIS and wAIS). The clear winner is wAIS. Note that the oracle policy $q_{\mu_*, \Sigma_*}$, which is not the optimal one (see section B.3 in the supplementary material), seems to give worse results than the the policy $q_{\mu_*, \Sigma_0}$, as wAIS with `sig_0` performs better than the "oracle" after some time.

In Figure 2, we examine 3 constant allocation policies given by $T = 50$ and $n_t = 2e3$; $T = 20$ and $n_t = 5e3$; $T = 5$ and $n_t = 2e4$. We clearly notice that the rate of convergence is influenced by the number of update steps (at least at the beginning). The results call for updating as soon as possible the sampling policy. This empirical evidence supports the theoretical framework studied in the paper which imposes no condition on the growth of $(n_t)$.

**Acknowledgments**

The authors are grateful to Rémi Bardenet for useful comments and additional references.

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
