[Supplementary Material]

# Supplementary material
# Asymptotic optimality of adaptive importance sampling

**Bernard Delyon**
IRMAR
University of Rennes 1
bernard.delyon@univ-rennes1.fr

**François Portier**
Télécom ParisTech
University of Paris-Saclay
francois.portier@gmail.com

To allow the reader to distinguish between the equations (or other numbered statements) of the main paper from the ones of the supplementary material, the equations (and other numbered statements) of the supplementary material are numbered as A.1, A.2, B.1, etc..., where the letter indicates the section.

## A    Proofs of the stated results

### A.1    Proof of Lemma 1

For the first statement, it suffices to note that $\mathbb{E}[\Delta_j|\mathscr{F}_{j-1}] = 0$ because $x_j$ is drawn according to $q_{j-1}$. For the second statement, the conditional independence implies that

$$\mathbb{E}\big[\Delta_j\Delta_j^T \,|\, \mathscr{F}_{j-1}\big] = V(q_{j-1}, \varphi).$$

$\square$

### A.2    Proof of Theorem 1

We need to show that for each $\gamma \in \mathbb{R}^p$, $\langle \sqrt{n}(I_n - \int \varphi), \gamma \rangle \xrightarrow{\mathrm{d}} \mathcal{N}(0, \gamma^T V_* \gamma)$. This reduces the proof to the case where $\varphi$ is a real-valued function, which is assumed below.

Since $\sqrt{n}\,(I_n - \int \varphi) = n^{-1/2}M_n$, this theorem will be a consequence of Corollary 3.1 p. 58 in [2] if we can prove that the martingale increments

$$X_{n,j} = \frac{1}{\sqrt{n}}\Delta_j$$

satisfy the following two conditions:

$$\sum_{j=1}^{n} \mathbb{E}[X_{n,j}^2|\mathscr{F}_{j-1}] \to V_*, \ \text{ in probability,} \tag{A.1}$$

$$\forall \varepsilon > 0, \ \ \sum_{j=1}^{n} \mathbb{E}[X_{n,j}^2 1_{|X_{n,j}|>\varepsilon}|\mathscr{F}_{j-1}] \to 0, \ \text{ in probability.} \tag{A.2}$$

Reformulating Proposition 1, we get

$$\sum_{j=1}^{n} \mathbb{E}[X_{n,j}^2|\mathscr{F}_{j-1}] = n^{-1}\langle M\rangle_n = V_* + n^{-1}\sum_{j=1}^{n}(V(q_{j-1}, \varphi) - V_*).$$

By the Cesaro Lemma, using (4), the right term in the previous display goes to 0 a.s., i.e., $n^{-1}\langle M\rangle_n \to V_*$.

Concerning (A.2), we have

$$\sum_{j=1}^{n} \mathbb{E}[X_{n,j}^2 1_{|X_{n,j}|>\varepsilon}|\mathscr{F}_{j-1}] = \frac{1}{n}\sum_{j=1}^{n}\mathbb{E}[\Delta_j^2 1_{|\Delta_j|>\varepsilon\sqrt{n}}|\mathscr{F}_{j-1}]. \tag{A.3}$$

Let us recall that

$$\Delta_j = w_j(x_j) - \int \varphi,$$

$$w_j(x) = \frac{\varphi(x)}{q_{j-1}(x)},$$

and introduce $I = \int \varphi$. Thus

$$\mathbb{E}[\Delta_j^2 1_{|\Delta_j|>\varepsilon\sqrt{n}}|\mathscr{F}_{j-1}]$$

$$= \int (w_j(x) - I)^2 1_{\{|w_j(x)-I|>\varepsilon\sqrt{n}\}} q_{j-1}(x)dx$$

$$\leqslant \int 2(w_j(x)^2 + I^2) 1_{\{|w_j(x)|>\varepsilon\sqrt{n}-|I|\}} q_{j-1}(x)dx$$

$$= 2\int \frac{\varphi(x)^2}{q_{j-1}(x)} 1_{\{|w_j(x)|>\varepsilon\sqrt{n}-|I|\}} dx + 2I^2 \int 1_{\{|w_j(x)|>\varepsilon\sqrt{n}-|I|\}} q_{j-1}(x)dx.$$

Let $\eta > 0$. Assuming that $\sqrt{n} > |I|/\epsilon$ and applying 2 times Markov inequality we obtain that

$$\mathbb{E}[\Delta_j^2 1_{|\Delta_j|>\varepsilon\sqrt{n}}|\mathscr{F}_{j-1}]$$

$$\leqslant \frac{2}{(\varepsilon\sqrt{n}-|I|)^\eta} \int \frac{|\varphi(x)|^{2+\eta}}{q_{j-1}(x)^{1+\eta}} dx + \frac{2}{(\varepsilon\sqrt{n}-|I|)} I^2 \int \left|\frac{\varphi(x)}{q_{j-1}(x)}\right| q_{j-1}(x)dx$$

$$\leqslant \frac{2}{(\varepsilon\sqrt{n}-|I|)^\eta} \sup_{j\in\mathbb{N}} \int \frac{|\varphi(x)|^{2+\eta}}{q_j(x)^{1+\eta}} dx + \frac{2|I|^2}{\varepsilon\sqrt{n}-|I|} \int |\varphi(x)|dx$$

which together with (A.3) implies (A.2). $\qquad\square$

### A.3  Proof of Corollary 1

Set

$$A_n = I_n(\varphi\pi) - \int \varphi\pi,$$

$$B_n = I_n(\pi) - 1.$$

We kwon by Theorem 1 that

$$\sqrt{n}\begin{pmatrix} A_n \\ B_n \end{pmatrix} \xrightarrow{\mathrm{d}} \mathcal{N}(0, V_*). \tag{A.4}$$

On the other hand, one has

$$I_n^{(\mathrm{norm})} - \int \varphi\pi = \frac{A_n + \int \varphi\pi}{B_n + 1} - \int \varphi\pi = \frac{A_n - B_n \int \varphi\pi}{B_n + 1} = \frac{1}{B_n + 1} U\begin{pmatrix} A_n \\ B_n \end{pmatrix},$$

where $U$ is the matrix in defined Corollary 1. The result then follows immediately from (A.4) and Slutsky's Lemma [3, chapter 2]. $\qquad\square$

### A.4  Proof of Theorem 2

Following [3, Theorem 5.7], we just need to show that

$$\sup_{\theta\in\Theta} |n^{-1}R_n(\theta) - r(\theta)| \to 0 \quad \text{a.s.} \tag{A.5}$$

$$\forall\varepsilon > 0, \quad \inf_{\theta\in\Theta,\, \|\theta-\theta_*\|>\epsilon} \int m_\theta > \int m_{\theta_*}. \tag{A.6}$$

Since $\Theta$ is compact, the second equation is satisfied because the integrability of $M$ implies, by the Lebesgue theorem, that the function $\theta \mapsto \int m_\theta$ is contituous.

Concerning (A.5), we shall apply Theorem B.1 (given in Section B.1 of the present supplementary material) with

$$H(\theta) = H(\theta, \omega) = \frac{m_\theta(X)}{q_0(X)}, \quad \text{where } X \sim q_0$$

$$H_j(\theta) = H_j(\theta, \omega) = \frac{m_\theta(x_j)}{q_{j-1}(x_j)}.$$

The two assumptions to verify, (H1) and (H2), are stated in Section B.1. In fact, we only have to show that (B.1), (B.2) and (B.3), expressed in (H1), hold true as the continuity of $\theta \mapsto H(\theta, \omega)$ almost surely, for each $\theta \in \Theta$, required in (H2), is a consequence of the continuity of $\theta \mapsto m_\theta(x)$. Notice that we have indeed $\mathbb{E}[H(\theta)] = \int m_\theta$, as (11) implies that for each $\theta$, the support of $M$ is included in the support of $q_0$.

For (B.1), we apply Theorem B.2 (given in Section B.2 of the present supplementary material) firstly with $U_j = H_j(\theta_0)_+$. Since $\mathbb{E}[U_j | \mathscr{F}_{j-1}] = \int m_{\theta_0}(x)_+ dx$, we get

$$\mathbb{E}[S_n] = \int m_{\theta_0}(x)_+ dx,$$

and

$$\mathrm{Var}(S_n) = \sum_{j=1}^n \mathrm{Var}(U_j) \leqslant \sum_{j=1}^n \mathbb{E}[U_j^2],$$

where the previous equality follows from $\mathrm{Cov}(U_i, U_j) = 0$, for all $i < j$. But, for each $j$,

$$\mathbb{E}[U_j^2] = \mathbb{E}\left[\frac{m_{\theta_0}(x_j)^2}{q_{j-1}(x_j)^2}\right] = \mathbb{E}\int \frac{m_{\theta_0}(x)^2}{q_{j-1}(x)} dx \leqslant \sup_{\theta \in \Theta} \int \frac{m_{\theta_0}(x)^2}{q_\theta(x)} dx.$$

This proves that (B.1) holds with $H_j(\theta_0)_+$ instead of $H_j(\theta_0)$. But similarly it holds with $H_j(\theta_0)_-$ and we conclude for $H_j(\theta_0)$ by linearity. Now (B.2) reduces to

$$\int \sup_{\theta \in \Theta} |m_\theta(x)| dx < \infty$$

which is true by assumption. Concerning (B.3), we work similarly with

$$U_j = \sup_{\theta \in B} |H_j(\theta) - H_j(\theta_0)|.$$

Now

$$\mathbb{E}[S_n] = n \int \sup_{\theta \in B} |m_\theta(x) - m_{\theta_0}(x)| dx = \mathbb{E}\left[\sup_{\theta \in B} |H(\theta) - H(\theta_0)|\right],$$

and

$$\mathrm{Var}(S_n) = \sum_{j=1}^n \mathrm{Var}(U_j) \leqslant \sum_{j=1}^n \mathbb{E}[U_j^2],$$

with

$$\mathbb{E}[U_j^2] = \mathbb{E}\int \frac{\sup_{\theta \in B} |m_\theta(x) - m_{\theta_0}(x)|^2}{q_{j-1}(x)} dx \leqslant 2 \sup_{\theta \in \Theta} \int \frac{\sup_{\theta \in B} m_\theta(x)^2}{q_\theta(x)} dx.$$

This leads similarly to (B.3). $\qquad\square$

## A.5 Proof of Theorem 3

Note that $V(q, \varphi) = \int \varphi \varphi^T / q - \int \varphi \int \varphi^T$. Because of the uniform integrability of $\|\varphi\|^2 / q_\theta$ the map $\theta \mapsto V(q_\theta, \varphi)$ is continuous. Hence, in virtue of the continuous mapping theorem and the conclusion of Theorem 2, Condition (4) is satisfied. Condition (5) is trivially satisfied. $\qquad\square$

## A.6  Proof of Corollary 2

The proof is the same as the proof of Theorem 3 replacing $\varphi$ by $(\varphi^T \pi, \pi)^T$.  $\square$

# B  Auxiliary results

## B.1  A uniform law of large numbers

We consider a compact metric space $(\Theta, d)$, and a sequence of stochastic processes $H_i(\omega) = H_i(\theta, \omega) : \Omega \to \mathbb{R}^d$, $i \geqslant 1$, $\theta \in \Theta$, such that:

**(H1)** There exists a stochastic processes $H(\theta) = H(\theta, \omega)$, such that for all $\theta_0 \in \Theta$

$$\frac{1}{n} \sum_{i=1}^{n} H_i(\theta_0) \longrightarrow \mathbb{E}\big[H(\theta_0)\big] \quad \text{a.s.} \tag{B.1}$$

In addition

$$\mathbb{E}\Big[ \sup_{\theta \in \Theta} \big|H(\theta)\big| \Big] < \infty \tag{B.2}$$

and for any ball $B$ with center $\theta_0$

$$\frac{1}{n} \sum_{i=1}^{n} \sup_{\theta \in B} \big|H_i(\theta) - H_i(\theta_0)\big| \longrightarrow \mathbb{E}\Big[ \sup_{\theta \in B} \big|H(\theta) - H(\theta_0)\big| \Big] \quad \text{a.s.} \tag{B.3}$$

The measurability of the supremum is part of the assumptions.

**(H2)** For each $\theta_0 \in \Theta$, almost surely (this subset of $\Omega$ of probability 1 may depend on $\theta_0$), the function $\theta \mapsto H(\theta, \omega)$ is continuous at $\theta_0$.

**Theorem B.1.** (UNIFORM LAW OF LARGE NUMBERS) *Under (H1) and (H2), the function*

$$h(\theta) = \mathbb{E}\big[H(\theta)\big]$$

*is continuous and with probability 1*

$$\limsup_{n} \sup_{\theta \in \Theta} \Big| h(\theta) - \frac{1}{n} \sum_{i=1}^{n} H_i(\theta) \Big| = 0. \tag{B.4}$$

*Proof.* Let us consider, for any $\theta_0 \in \Theta$, the function

$$f_{\theta_0}(\eta) = \mathbb{E}\Big[ \sup_{d(\theta, \theta_0) < \eta} \big|H(\theta) - H(\theta_0)\big| \Big].$$

Then $f_{\theta_0}(\eta)$ tends to 0 as $\eta$ tends to 0, because of (H2) and Lebesgue's dominated convergence Theorem. This implies in particular the continuity of $h(\theta)$ since clearly

$$\sup_{d(\theta, \theta_0) < \eta} \big|h(\theta) - h(\theta_0)\big| = \sup_{d(\theta, \theta_0) < \eta} \big|\mathbb{E}[H(\theta) - H(\theta_0)]\big| \leqslant f_{\theta_0}(\eta).$$

Fix $\varepsilon > 0$. For any $\theta_0$, there exists $\eta(\theta_0) > 0$ such that $f_{\theta_0}(\eta(\theta_0)) < \varepsilon$. The open balls centered at $\theta \in \Theta$ with radius $\eta(\theta)$ form a covering of cover $\Theta$; by compacity, a finite sub-covering exists:

$$\Theta = \cup_{j=1}^{J} B_j, \quad B_j = \big\{ \theta : d(\theta, \theta_j) < \eta(\theta_j) \big\}.$$

For any $\theta \in \Theta$, consider $j = j(\theta)$ the smallest $j$ such that $\theta \in B_j$, and write:

$$\frac{1}{n} \sum_{i=1}^{n} H_i(\theta) - h(\theta) = \frac{1}{n} \sum_{i=1}^{n} \{H_i(\theta) - H_i(\theta_j)\} + \frac{1}{n} \sum_{i=1}^{n} \{H_i(\theta_j) - h(\theta_j)\} + (h(\theta_j) - h(\theta)).$$

These three terms are functions of $\theta$, and we need to bound the uniform norm of them, not forgetting that $j$ depends on $\theta$. The supremum of the third one is smaller that $\varepsilon$; the supremum of the second one $Z_n(J)$ tends to 0 as $n$ tends to infinity: $J$ depends on $\epsilon$ but is finite, hence there exists a set $A_\epsilon$

such that $\mathbb{P}(A_\epsilon) = 1$ and $\forall \omega \in A_\epsilon$, $Z_n(J) \to 0$. Then set $A = \cap_{k \geqslant 1} A_{1/k}$, it holds that $\forall \omega \in A$, $Z_n(J) \to 0$. The first term is the only difficult one; its uniform norm is smaller than:

$$\varphi_n = \sup_j \frac{1}{n} \sum_{i=1}^n \sup_{\theta \in B_j} \left| H_i(\theta) - H_i(\theta_j) \right|.$$

But with probability 1, by virtue of (B.3)

$$\lim_n \varphi_n = \sup_j f_{\theta_j}(\eta) \leqslant \varepsilon.$$

We have shown that the l.h.s. of (B.4) is asymptotically smaller than $2\varepsilon$; since $\varepsilon$ is arbitrary, it actually vanishes. $\qquad\square$

## B.2   A law of large numbers

We present here a simple way to obtain the law of large numbers. This will be used for checking (B.1) and (B.3).

**Theorem B.2.** *Let $U_n, n \geqslant 1$ be a sequence of random variables and $S_n = U_1 + U_2 + ...U_n$ such that:*

$$U_n \geqslant 0 \quad w.p.1$$
$$n^{-1}\mathbb{E}[S_n] \longrightarrow l$$
$$\mathrm{Var}(S_n) \leqslant cn$$

*for some real numbers $c \geqslant 0$ and $l \geqslant 0$, then*

$$\frac{S_n}{n} \longrightarrow l \quad w.p.1.$$

*Proof.* The trick in this proof is to first derive the result for $S_{n^2}/n^2$. Then a sandwich formula will permit to conclude for $S_n/n$. We have

$$\mathbb{E}\left[ \sum_n \left( \frac{S_{n^2} - \mathbb{E}[S_{n^2}]}{n^2} \right)^2 \right] \leqslant \sum_n \frac{c}{n^2} < \infty.$$

Thus

$$\sum_n \left( \frac{S_{n^2} - \mathbb{E}[S_{n^2}]}{n^2} \right)^2 \quad \text{is finite w.p.1,}$$

implying that $(S_{n^2} - \mathbb{E}[S_{n^2}])/n^2$ converges to zero, almost surely. Hence $S_{n^2}/n^2$ converges to $l$. Notice that if $n^2 \leqslant k \leqslant (n+1)^2$:

$$\frac{S_{n^2}}{n^2} \frac{n^2}{(n+1)^2} \leqslant \frac{S_k}{k} \leqslant \frac{S_{(n+1)^2}}{(n+1)^2} \frac{(n+1)^2}{n^2}$$

and since both side terms tend to $l$, the result is proved. $\qquad\square$

## B.3   Algebra related to the optimal policy for normalized AIS

Suppose that $p = 1$. We have

$$V(q, (\varphi\pi, \pi)^T) = \begin{pmatrix} \overline{\rho_1^2} - \bar{\rho}_1^2 & \overline{\rho_1\rho_2} - \bar{\rho}_1 \\ \overline{\rho_1\rho_2} - \bar{\rho}_1 & \overline{\rho_2^2} - 1 \end{pmatrix}, \quad \rho_1 = \frac{\varphi\pi}{q}, \; \rho_2 = \frac{\pi}{q},$$

where the bar means the expectation under $q(x)dx$. From Corollary 1 with $U = (1, -\bar{\rho}_1)$, the asymptotic variance is

$$\begin{aligned}
UV(q, (\varphi\pi, \pi)^T)U^T &= (\overline{\rho_1^2} - \bar{\rho}_1^2) + \bar{\rho}_1^2(\overline{\rho_2^2} - 1) - 2\bar{\rho}_1(\overline{\rho_1\rho_2} - \bar{\rho}_1) \\
&= \overline{\rho_1^2} + \bar{\rho}_1^2\overline{\rho_2^2} - 2\bar{\rho}_1\overline{\rho_1\rho_2} \\
&= \mathbb{E}_q[(\rho_1 - \rho_2\bar{\rho}_1)^2] \\
&= \mathbb{E}_q[\rho_2\rho_2(\rho_1/\rho_2 - \bar{\rho}_1)^2] \\
&= \int [q^{-1}\pi^2(\varphi - I)^2].
\end{aligned}$$

Using Theorem 6.5 in [1], we derive the optimal sampling policy as claimed in the introduction.

Figure C.1: From left-to-right and top-to-bottom $d = 2, 4, 8, 16$. AIS and wAIS are computed with $T = 5, 20, 50$, each with a constant allocation policy, resp. $n_t = 2e4, 5e3, 2e3$. Different options are considered for estimating the variance : `sig_1`, `sig_1/2`, `sig_0` (see in the text). Plotted is the logarithm of the MSE (computed for each method over 100 replicates) with respect to the number of requests to the integrand.

## C  Additional numerical illustrations

In the numerical experiments furnished in the paper, the family of sampling policy has a fixed variance. Now we update the sampling policy according to the mean and the variance.

As detailed in the paper, we wish to compute $\mu_* = \int x\phi_{\mu_*,\sigma_*}(x)dx$ where $\phi_{\mu,\sigma} : \mathbb{R}^d \to \mathbb{R}$ is the probability density of $\mathcal{N}(\mu, \sigma^2\mathcal{I}_d)$, $\mu_* = (5,\dots 5)^T \in \mathbb{R}^d$, $\sigma_* = 1$. In contrast with the situation described in the paper, the sampling policy is now chosen in the collection of multivariate Student distributions of degree $\nu = 3$ denoted by $\{q_{\mu,\Sigma} : \mu \in \mathbb{R}^d, \Sigma \in \mathbb{R}^{d\times d}\}$. The initial sampling policy is set as $\mu_0 = 0$ and $\Sigma_0 = \sigma_0\mathcal{I}_d(\nu - 2)/\nu$ with $\sigma_0 = 5$. The mean $\mu_t$ and the variance $\Sigma_t$ are updated at each stage $t = 1,\dots T$ following the GMM approach as described in section 3 of the paper, leading to the simple update formulas, (7) for $\mu_t$ and (8) for $\Sigma_t$, with $f = \phi_{\mu_*,\sigma_*}$ (quoted equations are given in the paper). The variance estimation will be tuned : (i) complete variance estimation as described by (8), refereed to as `sig_1`; (ii) estimation restricted to the diagonal with 0 elsewhere, refereed to as `sig_1/2`; (iii) and without estimating the variance at all, refereed to as `sig_0`. To avoid degeneracy of the variance estimation in (i) and (ii), we add $\sigma_0/\max(1, N_t^{(\text{eff})})^{1/2}$

in the diagonal of $\Sigma_t$, with $N_t^{(\text{eff})} = \sum_{t=1}^{t} \sum_{i=1}^{n_t} \phi_{\mu_*,\sigma_*}(x_{s,i})/q_{s-1}(x_{s,i})$. The method described in (iii), `sig_0`, is the one considered in the paper.

For each method that returns $\mu$, the mean square error (MSE) is computed as the average of $\|\mu - \mu_*\|^2$ computed over 100 replicates of $\mu$.

In Figure C.1, we compare the evolution of all the mentioned algorithms with respect to stages $t = 1, \ldots T = 50$ with constant allocation policy $n_t = 2e3$ (for AIS and wAIS). The clear winner is wAIS without estimating the variance `sig_0`. Estimating the variance from the beginning of the procedure is slowing down the convergence especially in high dimensions.