[Reviews · NeurIPS 2018]

Reviewer 1



The paper is about popular adaptive importance sampling (AIS) schemes. The authors formulate a fairly general central limit theorem (CLT) for the AIS, and also introduce an original 'weighted' AIS scheme, which is similar to the one suggested in [1]. The authors report some promising empirical findings for their weighted AIS. The central limit theorem relies on a (fairly obvious) martingale representation, stemming from the unbiasedness of importance sampling. The result is different from existing CLTs in that it does not assume the number of samples per sampling stage to increase. This might be appealing in certain applications. The required condition (V(q_n,\phi) -> V_* a.s.) is rather implicit, and it is unclear how this condition can be checked in practice. The weighted AIS is presented briefly, and tested empirically. It is noted in the abstract and the introduction, that samples may be 'forgot' by the weighted AIS, but I failed to recover from §4.1 how this happens in practice. There is not much theory to support the method. Estimation of variance can be misleading, and the authors do regularisation in their experiments. This could be discussed in the weighted AIS §4.1. It may be also misleading to write about 'efficiency,' without taking into account computational aspects (complexity of the algorithm). When using a direct implementation of the AIS, the cost may tend to infinity, if all samples are retained. In such a case, the algorithm is definitely not 'efficient,' as stated in the abstract. Minor remarks: * typo: "Liebler", should be "Leibler" (several times) * p.5 below (9) typo: "compare compare" * Normalised AIS §2.3 needs some attention: It does not make (at least immediate) sense to have a product of normal *distribution* and a constant u^T, and the definition of u is broken. * It is a misleading to call the 2001 paper [13] "state-of-the-art" (adaptive MCMC). [1] Douc, R., Guillin, A., Marin, J. M., & Robert, C. P. (2007). Minimum variance importance sampling via population monte carlo. ESAIM: Probability and Statistics, 11, 427-447.

Reviewer 2



In this manuscript the authors consider adaptive importance sampling (AIS) schemes for estimating the integral of some target function \phi. To estimate such an integral a probability distribution (called sampling policy) generates an i.i.d. data sample and the integral is estimated by the empirical average of the function \phi weighted by the sampling policy. The key ingredient in adaptive sampling is to do multiple iterations of this procedure, updating the sampling procedure between each iteration to optimize the estimation. The authors point to many convergence results regarding AIS but stress that these are in the case where the number of iterations is fixed while the number of samples taken per iteration goes to infinity. They argue that a more natural and general setting is where the total number of samples of all iterations goes to infinity, which includes the previous case as well as the case where the sample size per iteration stays fixed but the total number of iterations tends to infinity. In this setting the authors use Martingale techniques to prove a central limit theorem for both standard and normalized AIS. Both results are independent of the exact update strategy. Next the authors consider a specific class of sampling policies, where the sampling policy is picked from a parametrized set. They prove that under certain mild assumptions, by applying risk minimization one obtains a collection of sampling policies that will converge to the optimal sampling policy. From this they then deduce a central limit theorem for parametric sampling policies, using risk minimization. Finally, a weighted update scheme for sampling policies is introduced. The weights enable the exclusion of previous bad updates when sampling the new data points. Using numerical experiments the authors show that such weighted AIS perform very well in practice as well as display their results for AIS. This paper is well-written with a clear overall structure, well defined objects and clearly stated results. In particular, I appreciate the fact that the authors make specific reference to the exact results from other papers they use for their proofs, which make the proof checking so much easier. I agree with the authors that the setting where the total number of samples tends to infinity is much more natural and hence I think their results are significant. I do however have a small problem with the proof of Theorem 5, since it assumes independence between variables U_i and U_j which does not seem to hold. In addition, I felt that the section on experiments was a bit short, which makes it hard for other researchers to duplicate the results. Apart from these two point I only have some small comments which I have listed below. In summary, I think the result on adaptive importance sampling presented in this paper are relevant and new, and the numerical experiments clearly show that weighted AIS could improve estimations. Therefore I would recommend this paper for acceptance in NIPS. ADDED AFTER FEEDBACK: My main concern regarding the proof of Theorem 5 was properly addressed and the authors agreed to improve/expand the experiments section. My overall recommendation regarding this paper remains unchanged. Comments main text: Line 1, [policy of stage t]: Within the first six words you introduce a variable that is not defined. Please consider changing the text to be more high level. Line 8, [raised in the paper]: I would change this to: raised in this paper. Line 17, [choice of the sampling…]: I would write: choice of a sampling… Algorithm (AIS): This is just an abstract representation of an AIS scheme, since no explicit update scheme is provided. I would be helpful to the reader if this was stated explicitly in the text. Line 57, [is there a weighted…]: I would change this to: is then a weighted. Line 61, [based on parametric]: I would write: based on a parametric. Line 82, [inspired from the]: I would change this to: inspired by the. Algorithm AIS at sample scale: Again, this seems to be an abstract algorithm. Please specify this in the text. The for loop refers to i but the index in the rest of the algorithm is j. Which one is it? Also, you explain what x_j is but what about q_{j-1}? Line 106: Here you use I without ever specifying that it is \int \phi, which creates confusion with the reader. Please define this. Proposition 1 and 2: Given the short proofs and the fact that these are just simple technical statements I would recommend merging these into one lemma. Line 118: It would help the reader if you recall that \phi maps into \mathbb{R}^p. Line 130: I do not understand what you mean with the part: \int \phi \pi, not \int \phi. Could you explain this a bit more. Line 140: What does (\phi \pi, \pi) stand for? I thought we only needed to replace \phi with \phi \pi in (4) and (5). Section 3: Here there is a huge jump with respect to the previous part. In particular, \theta_t is never defined not mentioned in the previous text. This makes it very difficult for the reader to digest this part. Please provide a bit more introduction here. Expression below line 156: I thought we were using parametric sampling policies q_{\theta}. How are the q_{s-1} related to these? This is particularly important since they appear again in equation (10). Line 161: Here there is a typo, compute twice. Line 179-180: The expression given her for R_t(\theta) seems to be exactly the same as (10). Line 208: Please remind the reader that V(q,\phi) is defined as in (3). Line 230: How do (8) and (9) work in practice? I thought we did not know q^\ast. Could you please add more explanation so that other researcher can reproduce these experiments. Comments supplementary material: Line 13, [using (4)]: Please be explicit when referring to equations in the main text to avoid unnecessary confusion. Also it might help to provide a bit more details on the use of Cesaro, for instance state that a_n = \sum_{j = 1}^n (V(q_j,\phi) - V_\ast) and b_n = n. Derivation below line 18: At the end of the first line there should be absolute value bars around \frac{\phi(x)}{q_{j-1}(x)}. In addition the equation sign on the next line should be a \leq sign. Line 24, [apply Theorem 1]: This is very confusing since the main text contains no Theorem 1. Please be very specific that you are referring to results that you will discuss later in this text. Line 25-26, [(H1)…(H2)]: Similar to the above comment, please be very specific that both these refer to statement that appear later in the text. Or preferably, move section 2.2 before the proof of Theorem 5. Derivation below line 31: Here it seems that something is going wrong. The first equality sign is only true when U_i and U_j are independent, which I do not think they are since they involve the updated versions of the sampling policy which depends on previous samples. Theorem 2: I do not follow the proof, especially the use of the n^2. From the fact that Var(S_n) \le cn it immediately follows that E[(\frac{S_n – E[S_n]}{n})^2] goes to zero as n \to \infty. This then immediately implies concentration and hence the result. Also, given that this result is fairly straightforward, I would recommend changing this to a lemma.

Reviewer 3



This works considers the problem of importance sampling (IS), which is an important topic in multi-integration and Bayesian learning. I am not an expert on the area of importance sampling, but like to provide my comments based on my understanding of the paper: 1. The CLT established in Section 2 seems to be based on generating a sample point at a time. I think such a strategy would be the theoretical study more feasible, but seems to be not that practical in real use of IS. 2. Although the authors claim verifying the condition in Theorem, it is still not clear how to check this condition in the practical use of the proposed method. 3. The notation of this paper is not very clear. For example, the notation of m_{\theta} in eq (6) is not well defined. 4. The simulation study only considers the Gaussian case, which is not convincing since the Gaussian case is quite straight in IS. 5. There is no real data example to demonstrate the performance of the proposed method. 6. The figures contains too many curves, which makes readers difficult to get the message from the figures.